# Embryo, Relocation and Secondary Nests of the Invasive Species *Vespa velutina* in Galicia (NW Spain)

**DOI:** 10.3390/ani12202781

**Published:** 2022-10-15

**Authors:** Ana Diéguez-Antón, Olga Escuredo, María Carmen Seijo, María Shantal Rodríguez-Flores

**Affiliations:** Department of Vegetal Biology and Soil Sciences, University of Vigo, 32004 Ourense, Spain

**Keywords:** *Vespa velutina*, early males, nest size, nesting behavior, Galicia

## Abstract

**Simple Summary:**

*Vespa velutina nigrithorax* is an invasive species established in the European Union since 2004. Galicia (NW Spain) is one of the areas strongly affected by the invasion, with around 28,000 nests identified per year. In the area, when the weather conditions are suitable, the queen starts the life cycle by building the embryo nest and laying the firsts eggs. This first stage of the colony is composed of the queen, a few small workers and sometimes males, living in a fragile nest usually situated in a protected place. After this, the nest continues to develop to a larger size, which leaves the nest more exposed in places such as tree canopy. The period of decline begins in autumn with the appearance of breeding individuals (males and gynes) and ends with the fecundation of new queens that will form the future colony in the next cycle. The high reproduction rate of this species has led to the successful expansion of this species into many regions, such as Galicia. *Vespa velutina* is established in Galicia since 2012, causing significant losses in agriculture, beekeeping and being a risk for human health.

**Abstract:**

Invasive species become established in non-native areas due to their intrinsic characteristics and the ability to adapt to new environments. This work describes the characteristics of the nesting behavior of the invasive yellow-legged hornet (*Vespa velutina nigrithorax*) in Galicia (Northwest Spain). The first nest was detected in the area in 2012 and after that, the distribution pattern shows a species-invasion curve with slow progress at first but followed by rapid expansion. The nesting places for this hornet differ between the kinds of nests, while embryo nests are mainly found in buildings in spring, secondary nests are observed in vegetation in summer, autumn, and winter. The annual life cycle starts when the queen builds the embryo nests and starts to lay eggs. This leads to the emergence of the first workers, usually small in size, and sometimes a few males. After this stage, large nests called secondary nests are normally observed in most exposed sites. Relocation nests can also be observed; these are nests in the first stage of development presenting adults insects but without brood or meconium. The period of decline is characterized by the emergence of new queens and males, that are distinguishable even in the pupal stage, the appearance of two eggs per cell, and an irregular brood pattern.

## 1. Introduction

Most long-distance introductions of exotic species to new areas are the direct or indirect result of human activities with social and economic factors being as critical as biological factors [1]. *Vespa velutina nigrithorax* du Buysson, 1905 (Hymenoptera: Vespidae), commonly called the yellow-legged hornet, is an invasive species from Southeast Asia, probably from eastern China, which has become widespread in European countries presenting a high range impact on biodiversity and ecosystems [2,3,4,5]. There are several circumstances that can explain the successful invasion of non-native species. Firstly, these involve the individuals surviving the transit from one region to the other [6] and secondly, the adaptation of the *V. velutina* mated queens to the new environment [3,7]. Moreover, the potential sperm production of the males and the polyandry, which allow the mating of one queen with several males, are determining factors [8]. All these characteristics increase genetic diversity in the early stages of the invasion favoring the expansion [9].

The first sighting of *V. velutina* in the North of the Iberian Peninsula was in 2010 [10]. In a few years, the species colonized the Cantabric and Atlantic coasts [11]. The ability of the queens to fly very long distances [12], the ability of the species to adapt to new territories mainly along coastal margins [13,14], and the most probable simultaneous introduction from the north (on the Cantabric coast) and the southwest (on the Atlantic coast) were suggested as significant factors for the spreading.

Galicia (situated in NW Spain) is one of the geographical areas most affected by the presence of this invasive species in Spain. The area has an Atlantic climate characterized by subdued temperatures, abundant rainfall, mild winters and summers, and low annual thermal oscillation. Most of the territory (68.8%) is covered by forest land including properly arboreal species and to a lesser extent, shrubs. Eurosiberian vegetation characterizes the majority of the area, with deciduous trees, such as *Quercus robur*, *Q. pyrenaica*, *Castanea sativa*, and *Betula celtiberica*, as well as riparian species, such as *Alnus glutinosa*, *Salix atrocinerea*, *Fraxinus excelsior*, and *Acer pseudoplatanus*, or introduced perennial species, such as *Pinus pinaster*, *P. radiata*, *Eucalyptus globulus*, or *E. nitens*, among others, standing out. This type of vegetation provides emplacements for *V. velutina* nests making them difficult to detect. In this situation, more than 28,000 nests are being detected per season in recent years [15].

The impacts originated by *V. velutina* are difficult to quantify but some economic sectors are strongly affected. Damages can be observed mainly in agriculture and beekeeping, and the species also causes strong effects in ecosystems and forest activities, and it is a risk for human health as well [16]. Some authors mention that the competition for food resources and nest sites caused the displacement and losses of native biodiversity [17,18,19]. *V. velutina* is a generalist predator that feeds on a wide spectrum of animal organisms, including flies and other social wasps, but especially honey bees [20,21]. Furthermore, *V. velutina* acts as a vector for the transmission of different parasites and pathologies [22,23]. Beekeeping is the one of the sectors affected since hornets actively prey on a large number of honey bees, with bees being a major protein resource for the hornet larvae. The agricultural sector is also affected by the reduction of pollination services on crops mediated by honey bees or due to the direct effects of hornets on fruit trees [24]. The increase of *V. velutina* nests in both rural and urban areas keeps people on alert due to their health consequences causing social alarm [25,26]. In fact, society is increasingly turning to the relevant authorities for improving control measures, to promote investment in research about the species, and to adapt effective control methods.

The *V. velutina* life cycle is linked to meteorological conditions [14]. The queen emerges from its winter shelter and builds the embryo nest, and in a few days begins to lay eggs that originate the first adults. After that, either the size and population could increase beyond this embryo nest, or the workers build a new nest in a new location (relocation nest) [27]. In early summer, relocation nests are distinguishable because they are characterized by combs without eggs, larvae, pupae, or meconium and there are also many adults in them. In late summer, autumn, and winter, these nests increase in size and population, and are then classified as secondary nests. These kinds of nests can hold more than 13,000 hornets in late summer [28,29]. Then, the period of nest decline begins with the appearance of the males and the future queens ready for mating. After that, the mated queens go into hiding for the overwintering period [3].

Despite several papers being published about the structure of yellow-legged hornet’s secondary nests [29,30,31,32], there is still a lack of knowledge about *V. velutina* nests, particularly embryo nests. This work contributes to the understanding of the biological life cycle through the study of *V. velutina* nests collected in Galicia from the stages of embryo nest to secondary nest. Different types of yellow-legged hornet nests collected in the geographical area were studied regarding their size, structure, and population. Therefore, the aim is to determine the architecture of the nests by characterizing the size of their colonies and following the steps of nests construction and colony development.

## 2. Materials and Methods

### 2.1. Database of Vespa velutina Nests Detected in Galicia

The number of nests detected each year, and the emplacement of the nests was obtained from the *V. velutina* surveillance and control program in Galicia (Xunta de Galicia). This database is created with alerts given by citizens each year. It covers the period from when the first nests were detected (2012 year) to the end of 2021. The data were processed to obtain information on the date of the observation, location, and the place where the nests were found, distinguishing buildings or human constructions, vegetation, land, and beehives. Furthermore, vegetation was divided into deciduous trees, perennial trees, and shrubs when it was possible.

### 2.2. Nests Handling Procedures

A total of 110 nests from different parts of Galicia were collected and studied. The nests were carefully removed by hand to maintain their structure. These nests were stored in a freezer until analysis. Firstly, the nests were classified into embryo nests (62 nests) and secondary nests (44 nests). Furthermore, four nests were considered relocation nests. Date information, method of removal, altitude, place, and location were recorded.

### 2.3. Nest’s Structure Analysis

External structure was determined by measuring the perimeter, the height, and the number and position of the entrances. For the internal description, a longitudinal section was opened to count the number of combs and to extract them. The first comb was considered the first built. The following measurements were taken from each comb: comb area, number of cells per comb, mean area of the cells, and the cell-building rate per day.

The area of each comb was measured following the formula:$$CA_{n}=\pi *r1*r2$$
where *CA_n_* is the area of the comb, *r*1 is the radius of the largest diameter of the comb and *r*2 is the radius of the smallest diameter of the comb, and the sub-index *n* means the number of the measured comb.

In embryo nests, all the cells in each comb were counted, but in the case of secondary nests, squares of 9 cm^2^ along the diameter of the combs were drawn to count the cells. Then, the number of cells per comb was calculated using the formula:$$NC_{n}=\frac{CA_{n}*NC_{9}}{9}$$
where *NC_n_* is the total number of cells of each comb and *NC*_9_ is the mean number of cells counted in a square of 9 cm^2^.

The area of the cells was estimated by the formula:$$AC_{n}=\frac{CA_{n}}{NC_{n}}$$

Cell-building rate per day was estimated for embryo nests using the method proposed by [33]. This consists of dividing the number of recently built cells (i.e., empty and egg cells without meconium) found in a colony by the development of the egg stage.
$$C_{BR}=\frac{C_{e}}{E_{s}}$$
where *C_BR_* is the cell-building rate, *C_e_* is the empty and cell eggs without meconium, and *E_s_* is the mean length of the development of the egg state. *E_s_* could have a value of 13 during the queen phase before the first worker or male emerges and a value of 6 during the phase after the first worker or male emerges [33].

### 2.4. Nest Population

Nest population was estimated counting the total number of eggs, larvae, and meconium in each comb and the number of combs in each nest [34], according to the following formula applied for each comb:$$P_{n}=E_{n}+L_{n}+M_{n}$$
where *P_n_* is the population of the comb, *E_n_* is the number of eggs in the comb; *L_n_* is the larvae in a comb, and *M_n_* is the number of individuals that emerged from the comb. The accumulated population (*P_t_*) is the sum of the population calculated for each comb:$$P_{t}=P_{1}+P_{2}+\cdots +P_{n}$$

Other information referred to the quantity of meconium in the cells. A section of 9 cm^2^ was used to extract the meconium and was weighed separately. This was repeated on three parts of each comb to calculate the mean weight of the meconium of the comb, as the middle cells are used more often than the side cells. The number of hornets reared in each cell of the comb was used to calculate the quantity of meconium per cell. Hence, the estimation of the number of hornets emerging from each cell was based on the value of the mean weight of meconium, and similarly for a secondary nest (0.06 g) and for an embryo nest (0.04 g). The following formulas were used:$$T_{cs}=\frac{mM_{n}}{0.06}$$
$$T_{ce}=\frac{mM_{n}}{0.04}$$
where *T_cs_* is the number of hornets emerged from each cell in secondary nests, *T_ce_* is the number of hornets emerged from each cell in embryo nests, and *mM_n_* is the mean weight of the meconium. To calculate the hornets emerged from each comb in secondary and embryo nests (*Th_ns_* or *Th_ne_*, respectively), the following formulas were used:$$Th_{ns}=\frac{mM_{n}}{0.06}\times NC_{n}$$
$$Th_{ne}=\frac{mM_{n}}{0.04}\times NC_{n}$$

Furthermore, all hornets within the nest were also counted despite these not being representative of the total number of adults in the nests. These individuals were classified by caste (queen, worker, or male). The pupae were counted and examined and when the size of the cover was different, cells were opened, and if they were sufficiently developed for differentiation, the pupae were classified into gynes, workers, or males. In the case of the embryo nests, all the pupae were opened, even if there were no differences in sizes.

### 2.5. Statistical Analyses

Before statistical analyses, the variables were tested for normality using the Kolmogorov–Smirnov test. When the variables fulfilled the criterion of normality, one-way analysis of variance (ANOVA) was used to analyze differences between the nest structure and population variables. Significance was marked at α < 0.05. Dunnett’s test was used as post hoc method to distinguish significant differences between seasons. When the variables did not fulfil the criterion of normality a Kruskal–Wallis non-parametric test was done. The Statistical Package for the Social Sciences (IBM SPSS Statistics 24) software was used for the statistical analyses.

## 3. Results

### 3.1. Vespa velutina Nest Locations in Galicia

More than 154,000 *V. velutina* nests were reported to the surveillance and control program of the local government authorities in the period from December 2012 to December 2021. The number of nests has increased exponentially from 2012 to 2018. In 2019 there was a slight reduction but in 2020 and 2021 more than 28,300 nests each were reported by citizens (Figure 1).

Nowadays, the yellow-legged hornet is present throughout the Galician territory but 59% of the total nests were from altitudes below 200 m, 25% were found at altitudes between 200 and 400 m, and 16% at altitudes above 400 m. The provinces in which more nests were reported were A Coruña (76,479 nests), accounting for 49.6% of the alerts, and Pontevedra (45,590 nests), accounting for 29.6% of notifications. These are the provinces with more kilometers of coastline and lower altitudes above sea level.

The nest sites were mostly located in vegetation (61%), and the rest in buildings (36%) and soil (3%). Most of the nests found in buildings were discovered in spring, coinciding with the development of the embryo nests and refers to sites such as manhole covers, roof areas, unused attics, walls, waste containers, chimneys, windows, balconies, or even beehives. On the other hand, the nests found in autumn and winter were located mostly in deciduous vegetation or perennial trees, or shrubs (Figure 2).

The most common deciduous trees were alder (*Alnus glutinosa*), poplar (*Populus*), birch (*Betula*), walnut (*Juglans*), chestnut (*Castanea*), oaks (*Quercus*), willows (*Salix*), and fruit trees, such as *Pyrus*, *Malus*, *Prunus*, or *Ficus carica*. Within the perennial trees, they were usually eucalyptus (mainly *Eucalyptus globulus*) and conifers, such as *Pinus*, *Cedrus*, *Picea*, and *Cupressus*. Furthermore, nests were detected in *Laurus nobilis*, *Ligustrum, Olea europaea*, and *Citrus*, among others.

### 3.2. Nesting Structure: Characteristics of Embryo Nests

A total of 62 embryo nests were collected and studied. The structure of the embryo nests starts with a single point of union that joins it to the surface, called the petiole. Around this point, the queen makes the first comb of hexagonal cells. This structure is promptly surrounded by a soft, loose, straw-like cover forming an envelope. Then new layers are added turning around each other in a spiral. The number of envelope sheets increases as the nest is constructed but at least two sheets are present. At the bottom, a hole of about 1.5 cm serves as an entrance. During this period, before the first workers or males emerge, the queen builds the nest alone having a mean cell-building rate of around 1 cell/day. When the first individuals emerge the cell-building rate increases until 3 cells/day.

The perimeter of the embryo nests varied between 8 and 33 cm, with a mean value of 21.8 cm. Nest height ranged from 3 to 10 cm, the mean being 7.6 cm. Internally, the nests had one or two combs supported by small petioles. For the first comb, the area varied between 2 and 66 cm^2^ with a mean value of 14 cm^2^, and for the second comb, the size was between 2 and 24 cm^2^ with a mean value of 11.5 cm^2^ (Table 1).

### 3.3. Nesting Structure: Characteristics of Relocation Nests

Four nests removed in May were classified as relocation nests. These nests are characterized because they were found in early summer and the combs were empty of brood or meconium. The number of combs in these nests varied from one to three, however, one of them presented the external structure without any comb inside. The number of individuals within these nests was around 50 female hornets. Eggs but no meconium, larvae or pupae were observed in the combs. This observation suggests that the adults found inside of the nest came from a different nest. In the nest without combs it was possible to identify two queens, while in the other three nests of this type there was only one. These nests presented an external structure like that of the secondary nests of the earlier seasons considering that are the initial stage of the secondary nests.

### 3.4. Nesting Structure: Characteristics of Secondary Nests

A total of 44 secondary nests were collected and studied. Secondary nests had a consistent outer cover composed of compacted sheets. Frequently this cover is more than five centimeters wide. The color varies from light straw to dark grey. Nests had one main big lateral entrance and small secondary entrances situated from the middle to the bottom of the developed nest. The internal structure is formed by the combs joined with petioles. Parameters such as perimeter, height, combs, nests cells and cells area were determined (Table 1).

Secondary nests were from different periods of the life cycle, thus they had different development stages. The smallest one had a perimeter of 26 cm and a height of 8 cm, while the biggest had 162 cm perimeter and 120 cm height. Regarding the number of combs, this varied from only two combs to 11 combs, with a mean number of five combs. The smallest nest had only 33 cells, while 21,883 cells were calculated for the largest nest. The mean number of cells was 4805 with a standard deviation between nests related to the size of the nests. Significant differences were found between nests collected in spring and summer, and nests removed in autumn and winter, in terms of perimeter and height. The nests grow larger as the season progresses, thus the number of combs and therefore the number of cells increase continuously. Larger nests collected in autumn and winter contained thousands of cells (mean values of 8043 and 6593 cells, respectively). Early spring nests and the nests at altitudes above 400 m had the smallest cell area. On the other hand, autumn and winter nests showed higher cell area values (mean values 0.54 cm and 0.56 cm, respectively). The area of the cells varied from a mean of 0.39 cm^2^ for spring nests to a mean value of 0.56 cm^2^ for cells of winter nests. The cell area determines the size of the hornets, therefore hornets from smaller nests, including embryo nests, are smaller than hornets emerging from secondary nests.

Concerning the shape of the combs, it follows the pattern of the nest shape, adapted the size and shape to the presence of elements such as tree branches, or other structures used to support the nest. Normally, from the first comb (that often has a value over 250 cm^2^) to fourth comb, the area increases progressively. Comb five has the highest variability in size and from comb six to the last comb (depending on the nest size), the area decreases. The smallest nests are spherical in shape, but as they get larger, they become more oval.

Nests grow in width and height during the whole growing season. Figure 3 shows relationships between the perimeter and height (in logarithmic scale), and the month of the year. The perimeter increased until the end of October or the beginning of November (Figure 3a), while the height presented higher values until the month of December (Figure 3b).

### 3.5. Nesting Population Characteristics of the Different Types of Nests

All the embryo nests, except one, had some caste of hornet, normally workers and the queen. In eight there were queens, workers, and males. Despite finding two queens in one embryo nest, there were mainly only one queen and brood in different stages. The presence of males was demonstrated in embryo nests removed in April, May, and June. Each embryo nest with males also had workers. Populations in secondary nests increases exponentially from comb one to comb four, which is related to the first period of nest growing (with an accumulated mean value of 17,000 hornets). From comb five onwards, population increase is lower (Figure 4).

The lower the number of combs the higher the number of insects that emerge; therefore, combs have more layers of meconium as they are built earlier. As the number of combs increases, fewer individuals emerge from those combs.

The secondary nests are characterized by a mature mated queen and many workers. The brood pattern followed a series of regular concentric circles in all the combs. Eggs, larvae, and pupae were observed in all the combs (Figure 5).

When the period of decline starts, a scattered brood pattern can be observed. This irregular brood pattern, the emergence of the males and the new queens, and the appearance of two eggs in one cell (Figure 6), characterized the decline period. This was observed in nests removed from September to the winter period. The presence of two eggs in the same cell in embryo nests was observed as well.

This study exposes for the first time the finding of morphological differences between the size of the pupae and the presence of future queens (gynes) (marked with yellow dots), workers (with red dots), and males (with blue dots) (Figure 7).

The cells of future queens are the biggest ones, and the pupae covers protrude from the comb in width and height. In contrast, the male pupae covers are flush with the cell without protruding. In the case of the covers of the pupae of the workers, they protrude less than those of the future queens.

### 3.6. Estimation of the Population by Meconium Content

Meconium indicates that hornets were reared in the combs. More than one layer of meconium was observed in cells both in embryo and secondary nests. The weight of meconium in the embryo nest was (40 ± 4 mg, mean value) lower than meconium in secondary nest (60 ± 3 mg, mean value). In the middle of the first combs of secondary nests, the cells are used up to three times and in embryo nests, it was computed up to two times. The times the cells are used is longer for the first combs (until comb four) but are diminished from this. It was found that from comb eight to the last comb cells were used only once (Figure 8).

To estimate nest population, the total number of eggs, larvae, and the calculated emerged hornets were considered. There were significant differences in the content of larvae between the spring nests and the rest of the nests from the other seasons. The laying capacity of the queen is lowest at the beginning of the life cycle and at the end of the life cycle (winter). Maximum population is reached in autumn and winter as more insects are being born (Table 2).

## 4. Discussion

*Vespa velutina* is the unique hornet listed as an invasive species of concern by the European Union. It was first detected in France in 2004 [3] coming from Eastern Asia. The control and management measures carried out were not effective and now the species is abundant in several European countries. In these areas, the species is well known by society, causing concern. For this reason, research on this species, its biology, and its impacts has increased considerably. This work focuses on the study of the nests of the species, especially the embryo nests, contributing to the knowledge of its life cycle.

### 4.1. Nesting Habitat

Some factors affect the establishment of *V. velutina* colonies and the settling of nests, among them, meteorological parameters, and altitude or resource availability for nesting, food sources, or in combination [4,35,36]. Since the first nests were detected in Galicia (December 2012), the colony distribution pattern showed low progress at the beginning followed by a later exponential expansion. The nests have been appearing at higher altitudes as the hornets have become more established in the territory. Nests were first detected at very low altitude (no more than 100 m.a.s.l) but as the invasion progressed, nests appeared at higher altitudes inland, a few nests were even detected over 1700 m.a.s.l, in different years. The altitude above sea level plays an important role in the appearance of early colonies. Most of the nests (84%) were found at altitudes below 400 m and preferably near the coast. Although Robinet et al. [37] and Carvalho et al. [38] said that the presence of topographic barriers represents a constraint to *V. velutina* spread, it has been seen in inland regions of Galicia that hornets are also settling at higher altitudes than expected. Several nests were found at altitudes over 1700 m.

Concerning the emplacement, embryo nests were detected mainly in habited areas, in buildings, or human activity related sites such as in its native areas Taiwan [39] and Thailand [40], and the invasive areas such as South Korea [17] and France [29]. This ability to use protected sites to construct nests in early spring, allows them to be protected in adverse circumstances when the nests are more vulnerable to climatic conditions due to the fragility and weakness of the cover as occurs with other species such as *V. crabro* and *Dolichovespula saxonica* [41]. However, high positions in trees provide secondary nests a safe place to live and probably the raw material needed to form nest cover, without the need to move long distances [42,43,44]. Secondary nests were found mainly in forested areas (59.1%; trees and shrubs), that provide a hiding place in which they are easily camouflaged. The locations of the establishment of the colony are similar in other invasive areas such as South Korea [17], Portugal [38], or France [29]. Special attention should be given to those nests that are in shrubs and on the ground (5.1% of the total reported nests) because they are a risk for human activities, especially for forestry workers and farmers when cutting and clearing. Nest distribution in Galicia is recorded with the alerts given by citizens to the surveillance and control program for *V. velutina* in Galicia, hence the tendency to overestimate the nests in urban and inhabited areas in comparison to forest and non-inhabited areas should be considered. People have also become accustomed to living with this species, reducing the number of nest warnings. Socioeconomic, political, and administrative factors may affect nest recording in different places. Anyway, considering the high number of alerts each year, it is important to find useful techniques for detecting nests in the early stages of development, mainly in the tree canopy, where most of the secondary nests are located. Systems such as thermal imaging or radiotelemetry can be tested in different environments [45,46].

### 4.2. Life Cycle Dynamics and Nest Type

Despite the main stages of the *V. velutina* life cycle being well known, the timing of the different stages and the mechanisms that the yellow-legged hornet uses to create new colonies in non-native areas is still a challenge. The first embryo nests were detected in March, but the first observations of queens coincide with when the weather becomes mild after winter, frequently in February, while in South Korea the first queens searching for sap and nest material were observed in May [17]. These queens start the biological cycle by building the embryo nests at a cell-building rate mean of 1 cell/day and one comb to rear the first workers in [29,34,47]. The cell area determines which size the new hornet will have. The cells of these embryo combs have small size, thus the first workers and first males found in embryo nests were smaller than those found in secondary nests, as was observed in France [29].

The queen starts laying eggs in the middle of the comb and follows a regular circular pattern throughout the sides of the combs depending on the colony’s degree of development [31]. The length of time to develop from an egg to a worker in a medium-sized hornet such as *V. velutina* drops from around 50 days in the embryo nest to just 29 days in a mature colony as Prezoto et al. [9] who described the length of the cycle to be 33–53 days [48]. It could explain the long period from the time the queen emerges from overwintering (February) until the population peak in autumn when *V. velutina* causes maximum damage to apiaries [21].

Commonly, embryo nests had only one queen, however, two queens were found in four embryo nests in a form of cooperation or competition. Especially, when nests are destroyed it was mentioned for other species that the surviving queen may occupy other nests seeking to usurp the colony and killing the queen, as Kumano and Kasuya described [49]. However, two queens laying eggs offers a better survival strategy for small nests against predators [9]. Furthermore, despite the most significant number of males being found in autumn, several males were also detected early in the embryo nests both as adults and pupae. All these nests with early males also had females, thus it is expected these hornets are not from an unmated queen. The role of the early males in *V. velutina* colonies is unclear, but the possibility of mating with overwintering virgin reproductive females (gynes) has been mentioned [47]. In this study, whether these males were diploid or haploid was not verified. However, the predominance of diploid males was related to a loss of allelic diversity and fitness as Arca et al. [7] described. They are usually sterile and when produced early do not contribute to the colony growth as workers do [50]. Although the emergence of males in early spring in introduced species can cause founder effects and genetic drift reducing the genetic diversity, multiple mating can mitigate this effect [47].

Arca et al. [7] indicated that queens mate with a mean of 2.44–4.11 males. More research about the relative presence of diploid and haploid males in colonies and their role in the reproductive strategies of the species should be addressed.

The embryo nest period for the colony is the most vulnerable to predators since the queen needs to collect food to feed larvae and wood pulp to enlarge the nest, thus spending long periods of time away [27]. It was reported that a significant number of nests fail in this stage [47]. The cumulative sum of secondary nest population showed that one nest with more than nine combs can rear over 23,000 insects during the life cycle following a non-linear growth rate. This type of nest often appeared in October. The biggest nest measured had 11 combs with a total of 21,883 cells. A nest in Java was found with the same number of combs, 11, but with a smaller number of cells, 11,912 [51]. However, in Taiwan, nests could reach up to 20,000 cells [39]. Regarding the invasive areas, the biggest nest found in France had the same number of combs but 13,547 cells, less than in Galicia [29]. The structure of these secondary nests allowed the production of around 25,000 hornets. The population found is higher than those observed by Rome et al. [29] in France and by Herrera et al. [52] in Balearic Island.

Embryo nests can be enlarged or relocated as was observed in other places [29,40,53]. It could happen due to the lack of space to enlarge the nest or looking for a suitable and safer place to enlarge the colony. Moreover, when the removal techniques destroy the nest but not the hornets, those hornets could build a new nest. The manner used to move the colony from one nest to the other is not well known but the most accepted hypothesis is that the nests were losing population as time passed, thus the new colony was formed progressively not immediately [33]. Martin [27] mentioned that during nest relocation, several starter nests are often built and quickly abandoned before the workers settle in one adequate location. In a few days, a football-size nest was made, and the colony gradually moves to it.

As the hornets’ life cycle continues, workers are responsible for enlarging the nest along the vertical axis by adding new combs and along the horizontal axis by adding new cells to already established combs in secondary nests. First, combs of secondary nests and cells have the higher rate of reutilization, more meconium layers were found. These cells were situated especially around the central point of the combs and usually contain two or three meconia, whereas the cells studied in secondary nests from France were used up to four times [29]. This meconium layer is ejected from larvae before metamorphosis and deposited at the bottom of the cell and is not removed [9,27]. The last built combs of the nest are used mainly for reproductive individuals, males, and gynes and only one meconium was found in them as was described for *V. crabro* [41].

In autumn and winter, the nest continues actively even if there is more than one queen, thus group-level characteristics are not strongly perturbed by queen loss [3,54]. This period is called the nest decline period and it is characterized by the appearance of two eggs per cell, the emergence of mature males, and the new foundresses. The reproductive strategy of hornets is based on the synchronicity between sexually mature males and foundress emergence due to males only being able to transfer sperm ten days after emergence [8]. During the decline period it is supposed that the queen founder has died [9], or it is too old to maintain the level of pheromones in the entire nest [55]. The presence of two eggs in a single cell is common in other insects, such as honey bees, but in *V. velutina* this disorganized oviposition behavior was attributed to orphan nests that are more frequent at the end of the summer season [29].

During the decline stage, nest population decreases, and the new foundresses abandon the nest and may disperse into the region to hibernate in small cavities. In December and January nests were found with eggs, brood in different stages, workers, and males; however, in that period, it was considered that workers were too few to effectively feed the whole brood and they focused on overfeeding the sexual brood [56]. The foragers stop bringing food and the workers have nothing left to feed the larvae, thus many larvae are expelled as also occurs in the *Vespa orientalis* species [33,56,57]. It explains why hornets with larvae in their mouths were observed. *V. velutina* has successfully stabilized its population and adapted its life cycle to new conditions. Far from being controlled, the species has achieved an excellent distribution, active secondary nests are frequently observed in January and in February, and it is already possible to observe new foundresses foraging in flora.

## 5. Conclusions

In conclusion, *Vespa velutina* is completely established in the study area, developing a significant number of nests each year. In this work, the nesting behaviors of the species were described for embryo nests and secondary nests providing data about the life cycle. More field studies are essential to increase knowledge about this invasive species and to improve management in invaded areas.

## Figures and Tables

**Figure 1 animals-12-02781-f001:**
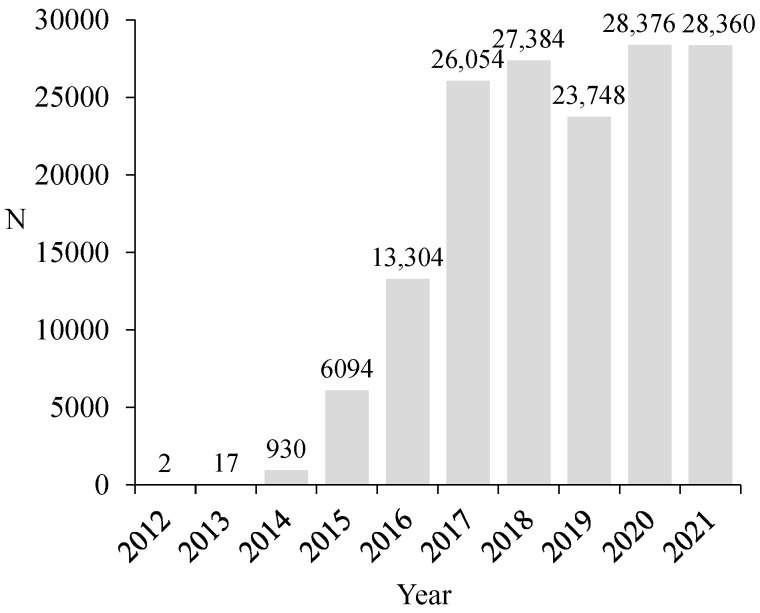
Total number of nests (N) reported from 2012 to 2021.

**Figure 2 animals-12-02781-f002:**
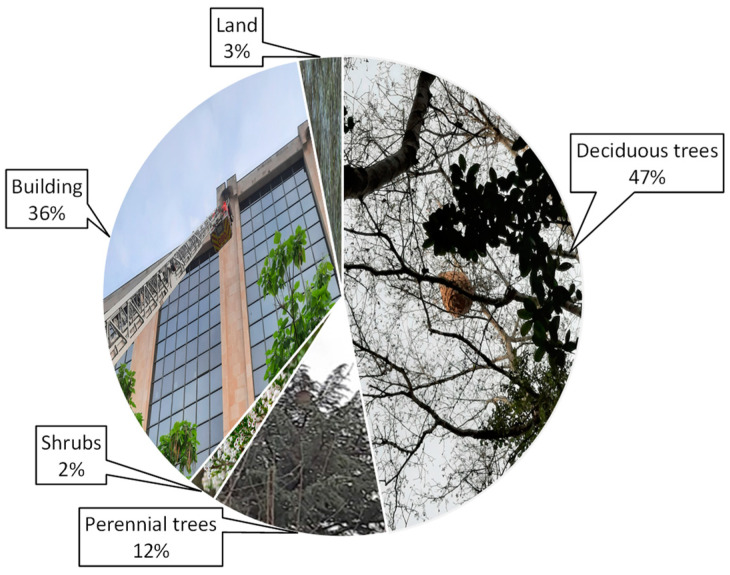
Classification of nests according to their location.

**Figure 3 animals-12-02781-f003:**
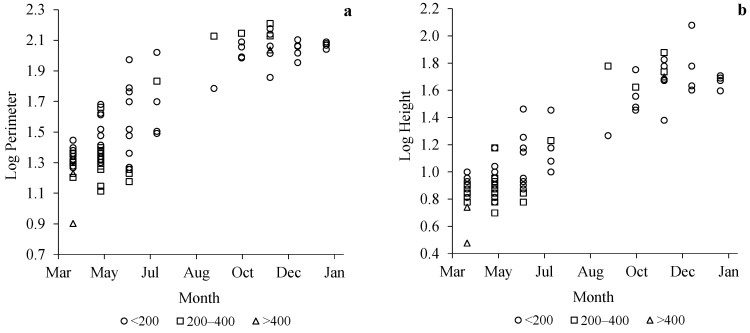
Variations in log perimeter (**a**) and log height (**b**) of nests collected per month classified according to altitudinal range (<200, 200–400, >400).

**Figure 4 animals-12-02781-f004:**
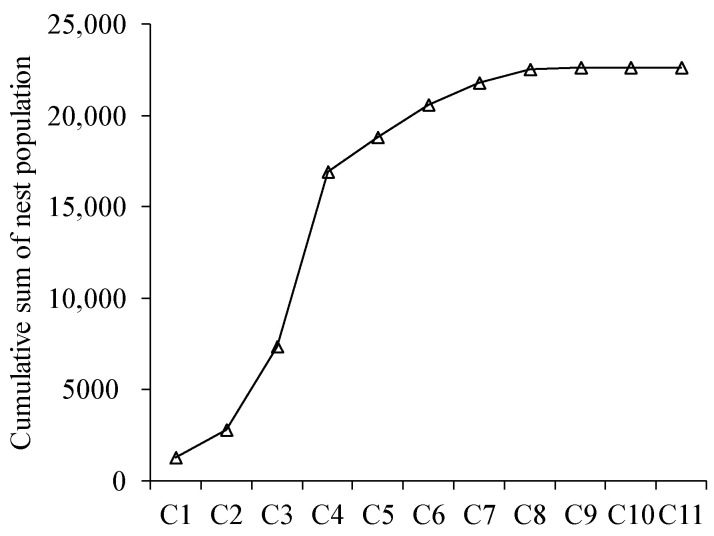
The cumulative sum of the mean population for each comb in secondary nests. From comb 1 (C1) to comb 11 (C11).

**Figure 5 animals-12-02781-f005:**
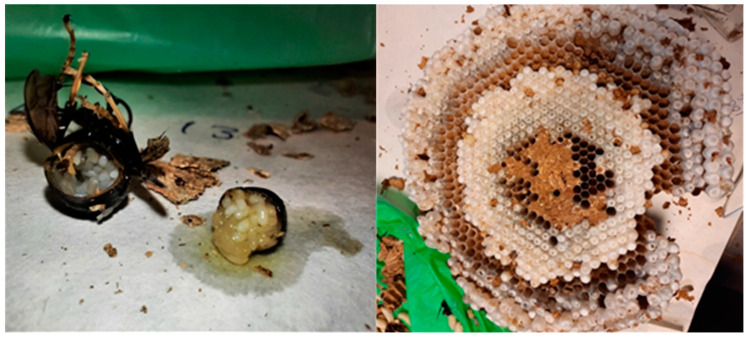
Mated queen (**left**) full of eggs and a comb with regular brood (**right**) in a mature secondary nest.

**Figure 6 animals-12-02781-f006:**
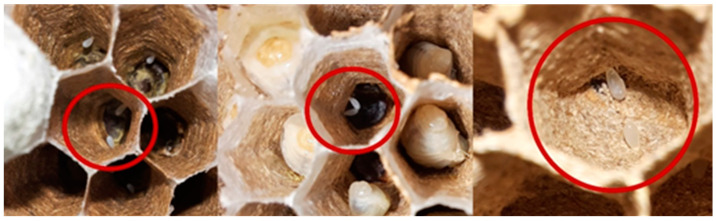
Two eggs per cell in different declining nests.

**Figure 7 animals-12-02781-f007:**
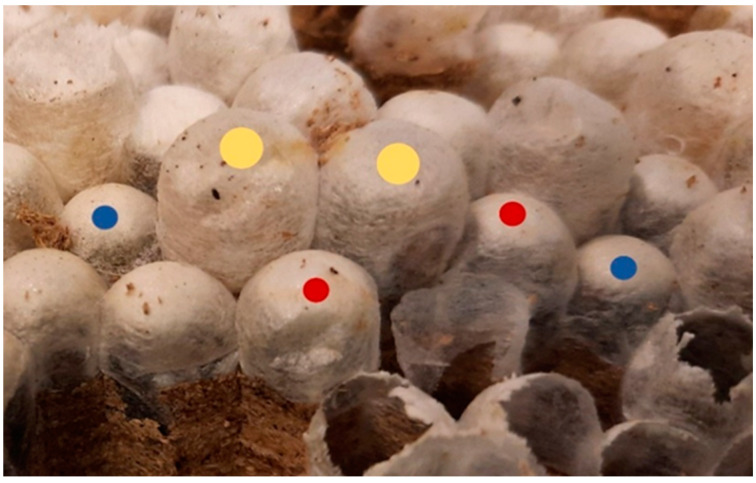
Morphological differences among the pupae size differing between future queens (marked with yellow points), workers (red points), and males (blue points).

**Figure 8 animals-12-02781-f008:**
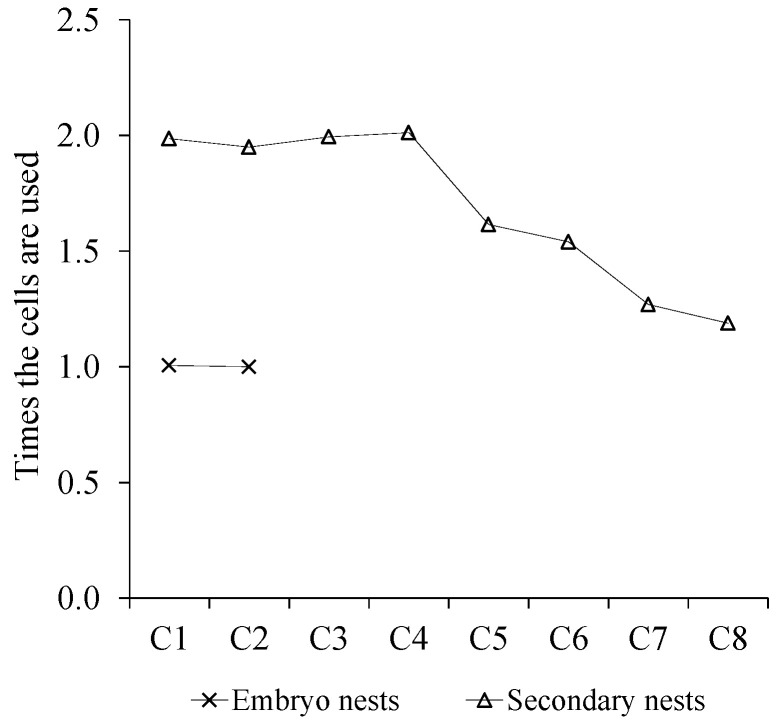
Mean of the times the cells are used in the different combs in embryo and secondary nests.

**Table 1 animals-12-02781-t001:** Mean ± standard deviation (SD) of structural characteristics of the different types of nests (embryo and secondary) were classified by season in which they were removed.

Nests Type	Season	N	Perimeter (cm)	Height (cm)	Combs	Nest Cells	Cells Area (cm^2^)
			Mean ± SD	Mean ± SD	Min-Max	Mean ± SD	Mean ± SD
Embryo	Spring	62	21.8 ± 4.8	7.6 ± 1.4	1–2	45 ± 33	0.31 ± 0.05
Secondary	Spring	7	43 ± 11.6 ^a^	12.8 ± 3.2 ^a^	2–4	575 ± 376 ^a^	0.39 ± 0.06 ^a^
	Summer	11	61.5 ± 26.8 ^a^	17.9 ± 7.1 ^a^	2–7	942 ± 1353 ^a^	0.47 ± 0.07 ^ab^
	Autumn	15	116.7 ± 27.9 ^b^	46.5 ± 16.5 ^b^	3–11	8043 ± 5920 ^b^	0.54 ± 0.06 ^bc^
	Winter	11	113.5 ± 11.1 ^b^	56.2 ± 26.6 ^b^	5–11	6593 ± 4005 ^b^	0.56 ± 0.03 ^c^
	All	44	94 ± 37	37.6 ± 23.5	2–11	4805 ± 5199	0.51 ± 0.08

Significant values are shown with different letters (*p* < 0.05).

**Table 2 animals-12-02781-t002:** Mean ± standard deviation (SD) of larvae, eggs, meconium larvae, and nest population of the different type of nests (embryo and secondary) classified by season when they were removed.

Nests type	Season	N	Larvae	Eggs	Hornets Emerged from Meconium	Nest Population
			Mean ± SD	Mean ± SD	Mean ± SD	Mean ± SD
Embryo	Spring	62	14 ± 12	12 ± 11	17 ± 20	32 ± 28
Secondary	Spring	7	158 ± 129 ^a^	112 ± 88 ^a^	369 ± 193 ^a^	85 ± 165 ^a^
	Summer	11	356 ± 505 ^a^	109 ± 125 ^a^	1347 ± 1135 ^ab^	1501± 1107 ^ab^
	Autumn	15	473 ± 416 ^a^	514 ± 10 ^b^	12,599 ± 10,479 ^c^	13,224 ± 10,561 ^b^
	Winter	11	298 ± 429 ^a^	101 ± 71 ^a^	12,145 ± 5984 ^bc^	12,255 ± 6040 ^ab^
	All	44	351 ± 407	162 ± 163	7643 ± 8746	8227 ± 8923

Significant values are shown with different letters (*p* < 0.05).

## Data Availability

All available data are incorporated in tables and figures within the manuscript.

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
