# Peer review of "Embryo, Relocation and Secondary Nests of the Invasive Species Vespa velutina in Galicia (NW Spain)"

_animals, 2022, doi:10.3390/ani12202781_

Round 1

Reviewer 1 Report

Dear Editor,

The authors of this study investigate nesting behavior of the invasive Asian hornet in Galicia (NW Spain). I have some suggestions to improve the submission.

Line 17. causing significant economic, social, and health impacts - who is negatively affected Vespa velutina?

Please, explain in the text.

Line 33-34. Please, avoid to repeating words mentioned in the title.

Line 39-40. invasive species – Please, clarify where the invasion of Europe is coming from.

Line 49. Do the authors have data on where this invasion is from?

Line 112. How these nests were collected?

Line 114. Also, four nests were considered relocation nests - by what criterion were these nests considered to be relocation nests? Please, give some more information.

Line 130. but in the case of secondary nests, squares of 9 cm2 along the diameter – why in secondary nest the measurement cell area is change?

Line 198. Please, mention that all variables are expressed as mean ± SD.

In Table 1 symbols a, b etc. should be in superscript.

In Figure 3 there is wrong legend – I think that is Figure 3a and 3b, instead of Figure 2a and 2b.

How many eggs are usually observed in the hornet cells?

Line 323. and the presence of future queens (gynes) - what are the signs to determine the future queens (gynes)?

Line 336. (40 mg, mean value) – any deviation from this value?

Line 349. Mean and standard deviation (SD) – I think that Mean ± standard deviation (SD) is more appropriate.

Line 355. Please, add some sentences about introducing of V. velutina in European countries, before the first part of the Discussion.

I think the publication would generate more reader interest if data were included on the negative impact of V. velutina in introduced areas, especially what the negative impact on the beekeeping sector is related to.

Reviewer 2 Report

Revision

Page 1, line 40: wich has became

References

Page 14, lines 521, 525: Vespa velutina

Page 14, line 589: du Buysson

Page 14, line 591: Provespa and Vespa

Page 14, lines 539, 544, 546: standardize the espaces after doi:

Page 16: line 612: (Vespinae, Hymenoptera)

 Legenda

In blue my suggestions, in red my corrections.

Reviewer 3 Report

Good and professional work.

I have found some small graphical defects in table legends, for ex. Fig. 8 (may be a problem of manuscript format or reader). 

Methodology is sufficient but can be improved for future experiments. I believe that comparing already killed nests with each other may not have such an informative value. It would be much more interesting to see progression of each nest during time. (I understand, that would be much more harder, or impossible in quarantine condition).
